# Scaling Large Language Model-based Multi-Agent Collaboration

**Chen Qian★†, Zihao Xie★†, YiFei Wang★, Wei Liu★, Kunlun Zhu★, Hanchen Xia★,**
**Yufan Dang★, Zhuoyun Du★, Weize Chen★, Cheng Yang♣, Zhiyuan Liu★, Maosong Sun★✉**
★Tsinghua University    ♣Peng Cheng Laboratory
qianc62@gmail.com    xie-zh22@mails.tsinghua.edu.cn    sms@tsinghua.edu.cn

## Abstract

Recent breakthroughs in large language model-driven *autonomous agents* have revealed that *multi-agent collaboration* often surpasses each individual through collective reasoning. Inspired by the neural scaling law—increasing neurons enhances performance, this study explores whether the continuous addition of collaborative agents can yield similar benefits. Technically, we utilize directed acyclic graphs to organize agents into a multi-agent collaboration network (MACNET), upon which their interactive reasoning is topologically orchestrated for autonomous task solving. Extensive evaluations reveal that it effectively supports collaboration among over a thousand agents, with irregular topologies outperforming regular ones. We also identify a *collaborative scaling law*—the overall performance follows a logistic growth pattern as agents scale, with collaborative emergence occurring earlier than traditional neural emergence. We speculate this may be because scaling agents catalyzes their multidimensional considerations during interactive reflection and refinement, thereby producing more comprehensive artifacts. The code is available at https://github.com/OpenBMB/ChatDev/tree/macnet.

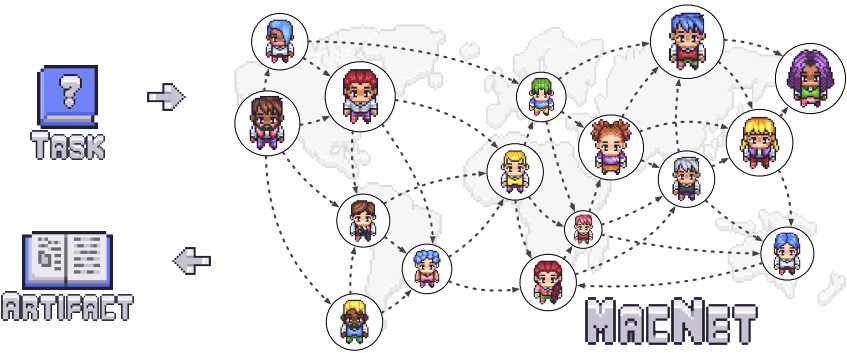

Figure 1: Multi-agent collaboration network (MACNET) uses directed acyclic graphs to arrange agents for collaborative interactions, facilitating autonomous task-solving through collective reasoning.

## 1 Introduction

In the rapidly advancing field of artificial intelligence, *large language models* (LLMs) have driven transformative shifts across numerous domains due to their remarkable linguistic capacity to seamlessly integrate extensive world knowledge (Vaswani et al., 2017; Brown et al., 2020). Central to this breakthrough is the *neural scaling law*, where well-trained neural networks often exhibit power-law scaling relations primarily with the number of neurons, alongside factors such as dataset size and training time (Kaplan et al., 2020; Muennighoff et al., 2024). Despite this, LLMs have inherent limitations in their enclosed reasoning, particularly when addressing complex situations that extend beyond textual boundaries (Schick et al., 2023). To this end, during the inference phase, pioneering

---

†: Equal Contributions.    ✉: Corresponding Authors.

studies transform foundational LLMs into versatile *autonomous agents* (Richards, 2023; Shen et al., 2023) by encapsulating external capabilities like context-aware memory (Park et al., 2023), tool use (Qin et al., 2024a), and procedural planning (Zhao et al., 2023). In this context, *multi-agent collaboration*, within an interactive environment, prompts agents to engage in iterative reflection and refinement, explicitly facilitating a process of "slow thinking" (Daniel, 2017; OpenAI, 2024). This paradigm effectively unites the distinct expertise of diverse agents (Qian et al., 2024c), ultimately leading to artifacts[1] derived from their dialogues.

Although numerous studies have confirmed that task-oriented multi-agent collaboration, facilitated by interactive behaviors, often surpasses standalone intelligence (Chen et al., 2024d;a), the potential for continuously increasing agents remains largely overlooked—with most research involving fewer than ten agents and only a limited number extending to several dozen (Li et al., 2023a; Park et al., 2023; Zhang et al., 2024a). Inspired by the neural scaling law, a thought-provoking question arises: *how does the continuous addition of collaborative agents impact performance?* Exploring the *collaborative scaling law* is essential for linking performance trends with inference resources, revealing underlying phenomena in agent networking, and promoting the development of scalable and predictable LLM systems. However, technically, effective collaboration should not depend on simple majority voting (Brown et al., 2024; Chen et al., 2024b); instead, it should incorporate strategic mechanisms for scalable networking, cooperative interaction, and progressive decision-making (Hopfield, 1982; Almaatouq et al., 2021; Du et al., 2024a). Toward this end, as depicted in Figure 1, we organize multiple agents into a multi-agent collaboration network (MACNET), upon which their interactive reasoning is topologically orchestrated for autonomous task solving.

- For network construction, agents' topology is constructed as a directed acyclic graph, with each edge managed by a supervisory *critic* issuing commands, and each node by a compliant *actor* providing tailored artifacts. This establishes a functional bipartition of labor among agents, promoting role specialization while inherently preventing backflow in information propagation.

- For interactive reasoning, agents interact in a topological order, where each round involves two adjacent agents refining a previous artifact, and only the refined artifact, rather than the entire dialogue, is propagated to the next rounds. This prevents global broadcasting and suppresses context explosion, thereby enhancing collaboration scalability for much larger networks.

We performed extensive evaluations across different downstream scenarios, employing three types of representative topologies—chain, tree, and graph—further divided into six representative variants. The results show that MACNET surpasses all baselines on average and supports effective collaboration among over a thousand agents. Counterintuitively, collaborating within irregular topologies unexpectedly outperforms that within regular ones. Notably, we reveal a *collaborative scaling law*, indicating that the overall performance exhibits a logistic growth pattern as the process of scaling agents, with collaborative emergence occurring earlier than previous instances of neural emergence. We speculate this may be because scaling agents catalyzes their multidimensional considerations during interactive reflection and refinement, thereby producing more comprehensive artifacts. Longer term, we aim for this research to extrapolate the traditional scaling from training to inference, circumventing the need for resource-intensive retraining through inference-time procedural thinking.

## 2 MULTI-AGENT COLLABORATION NETWORK

To create a scalable environment for effective collaboration, as depicted in Figure 1, we organize multiple agents into a multi-agent collaboration network (MACNET), upon which their interactive reasoning is topologically orchestrated for autonomous task solving.

### 2.1 NETWORK CONSTRUCTION

Although training-time neuron collaboration has been well-established with Transformer architectures (Vaswani et al., 2017), the suitable architectures for inference-time agent collaboration remain unclear and lack consensus. Toward this end, we draw on the concept of graphs—a data structure

---

[1]Artifacts can vary from multiple-choice answers to repository-level code or coherent narratives, among many other possibilities.

that describes entities and their interrelations—and extend from previous efforts to propose a more general topology as a *directed acyclic graph* (DAG) (Nilsson et al., 2020):

$$\mathcal{G} = (\mathcal{V}, \mathcal{E}) \quad \mathcal{V} = \{v_i | i \in I\} \quad \mathcal{E} = \{\langle v_i, v_j \rangle | i, j \in I \land i \neq j\} \tag{1}$$

where $\mathcal{V}$ denotes the set of nodes indexed by the index set $I$, and $\mathcal{E}$ denotes the set of edges, with each edge directed from one node to another and no cycles exist. A graph will orchestrate agent interactions, akin to social networks where information propagates through directed edges. Intuitively, the acyclic nature prevents information backflow, eliminating the need for additional designs like task-specific cycle-breaking, thereby enhancing generalizability and adaptability across contexts.

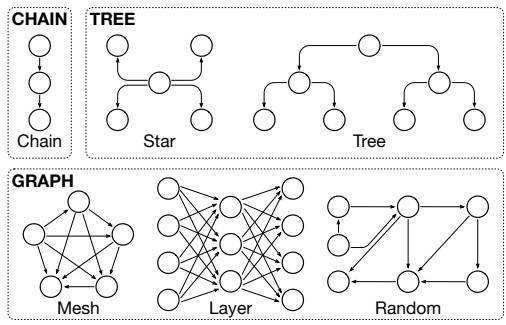

Figure 2: Representative topologies.

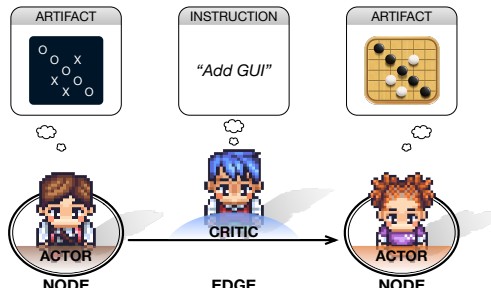

Figure 3: Assign functionally bipartite agents on nodes and edges, respectively.

Given the impracticality of enumerating all possible topologies, we focus on three prevalent types—chain, tree, and graph—further divided into six representative sub-topologies, as depicted in Figure 2. Chain topologies, resembling the waterfall model (Petersen et al., 2009), linearly structuring interactions along agents (Wei et al., 2022b; Hong et al., 2024). Tree topologies enable agents to branch out, interacting in independent directions (Yao et al., 2023; Zhuang et al., 2024); further categorized into "wider" star-shaped and "deeper" tree-shaped topologies. Graph topologies support arbitrary interaction dependencies, with nodes having multiple children and parents, forming either divergent or convergent interactions (Besta et al., 2024a; Chen et al., 2024d; Zhuge et al., 2024; Liu et al., 2023); further classified into fully-connected mesh topologies, MLP-shaped layered topologies, and irregular random topologies. These representative topologies are extensively studied in complex network (Dodds et al., 2003; Newman, 2001; Ma et al., 2024) and procedural reasoning (Zhang et al., 2024b; Yin et al., 2023; Besta et al., 2024b), ensuring a comprehensive coverage of the most widespread and practical topologies in multi-agent networking.

Since a functional bipartition—consisting of supervisory critics who issue directional instructions and compliant actors who provide tailored artifacts—can effectively establish division of labor, activate functional behaviors, and facilitate progressive task-solving (Li et al., 2023a), as depicted in Figure 3, we strategically assign a critic to each edge and an actor to each node:

$$\boldsymbol{a_i} = \rho(v_i), \ \forall v_i \in \mathcal{V} \quad \boldsymbol{a_{ij}} = \rho(\langle v_i, v_j \rangle), \ \forall \langle v_i, v_j \rangle \in \mathcal{E} \tag{2}$$

where $\rho(x)$ represents the *agentization* operation on an element $x$, achieved by equipping a foundation model with context-aware memory, external tools, and professional roles; $\boldsymbol{a_i}$ and $\boldsymbol{a_{ij}}$ denote an actor assigned to node $v_i$ and a critic assigned to edge $v_{ij}$, respectively.

## 2.2 INTERACTIVE REASONING

In procedural task-solving, interactive reasoning among agents within a static network requires strategical traversal to establish an orderly interaction criterion (Liu et al., 2024b; Chen et al., 2024e). In a directed acyclic setting, our graph traversal strategy adheres to the principles of *topological ordering* (Kahn, 1962), which ensures that each node is visited only after all its dependencies have been traversed. Formally, for a network $\mathcal{G}$, its topological order is a linear arrangement of agents $\boldsymbol{a_i}$ and $\boldsymbol{a_{ij}}$ such that for every directed edge $\langle v_i, v_j \rangle \in \mathcal{E}$, the ordering satisfies:

$$\forall \langle v_i, v_j \rangle \in \mathcal{E}, \ \mathbb{I}(\boldsymbol{a_i}) < \mathbb{I}(\boldsymbol{a_{ij}}) < \mathbb{I}(\boldsymbol{a_j}) \tag{3}$$

where $\mathbb{I}(x)$ denotes the index of agent $x$ in a topological sequence. This arrangement ensures that each node-occupied agent $\boldsymbol{a_i}$ precedes its corresponding edge-occupied agent $\boldsymbol{a_{ij}}$, and $\boldsymbol{a_{ij}}$ precedes $\boldsymbol{a_j}$, thereby ensuring orderly information propagation along the network.

Figure 4: Orchestrating the agents' reasoning process involves a series of dual-agent interactions. The topological order serves as the control flow, while the original connectivity governs the data flow.

After establishing the global order, as illustrated in Figure 4, we enable each pair of edge-connected adjacent agents to interact for artifact refinement, which results in a total assignment of $|\mathcal{V}| + |\mathcal{E}|$ agents and require at least $2 \times |\mathcal{E}|$ interaction rounds. Specifically, within each edge, the interactions between critics and actors follows a dual-agent multi-turn pattern:

$$\tau(\boldsymbol{a_i}, \boldsymbol{a_{ij}}, \boldsymbol{a_j}) = \big(\tau(\boldsymbol{a_i}, \boldsymbol{a_{ij}}), \tau(\boldsymbol{a_{ij}}, \boldsymbol{a_j})\big)$$
$$\tau(\boldsymbol{a_i}, \boldsymbol{a_{ij}}) = (\boldsymbol{a_i} \rightarrow \boldsymbol{a_{ij}}, \ \boldsymbol{a_{ij}} \rightsquigarrow \boldsymbol{a_i})_\circlearrowleft \quad \tau(\boldsymbol{a_{ij}}, \boldsymbol{a_j}) = (\boldsymbol{a_{ij}} \rightarrow \boldsymbol{a_j}, \ \boldsymbol{a_j} \rightsquigarrow \boldsymbol{a_{ij}})_\circlearrowleft \tag{4}$$

where $\tau(\cdot)$ represents the interaction between agents, $\rightarrow$ signifies an act of requesting, $\rightsquigarrow$ indicates a corresponding reply—within which the critic provides an instruction and the actor offers an artifact, and $\circlearrowleft$ denotes an iterative process. That is, $\boldsymbol{a_i}$ requests feedback, $\boldsymbol{a_{ij}}$ offers reflected suggestions and requests further refinement, and $\boldsymbol{a_j}$ provides a refined artifact. Thus, the agents associated with a single edge can engage in iterative reflection and refinement, effectively implementing an refinement of a previous artifact (Madaan et al., 2023; Renze & Guven, 2024).[2]

## 2.3 MEMORY CONTROL

Note that unrestrained information exchange among agents inevitably leads to *context explosion* (Liu et al., 2024b; Xu et al., 2024), ultimately hindering scalability by limiting support for additional entities. To address this, we adopt both short- and long-term memory to manage the context visibility for each agent (Sumers et al., 2023). *Short-term memory* captures the working memory within each interaction, ensuring context-aware decision-making (Li et al., 2023a). *Long-term memory* maintains context continuity by retaining only the final artifact derived from current dialogue, rather than the entire conversational history, ensuring that non-artifact contexts (*e.g.*, the detailed analysis process preceding an artifact) remain inaccessible[3] to subsequent agents (Qian et al., 2024c). This mechanism ensures that only the artifact propagates through the network, which explicitly minimizes context explosion risk while maintaining continuity. Artifacts propagate by branching at divergent nodes, or merging at convergent nodes requiring effective aggregation; technically, before refinement, convergent agents integrate the strengths of incoming artifacts through hierarchical aggregation (Du et al., 2024b) to yield a "non-linearly" strength-aggregated artifact.

Theoretically, in a mesh structure characterized by the highest interaction density, the total token consumption for the sink[4] agent who experiences maximum context pressure, with and without this mechanism, is derived as follows:

$$\mathcal{O}(n)_{\text{w/o}} = t + p + s + (2m - 1)(i + s)(n(n - 1)/2 + 2(n - 2)) \overset{n \gg 1}{\approx} Cn^2 \propto n^2$$
$$\mathcal{O}(n)_{\text{w/}} = t + p + s + m(i + s)((n - 1) + 2(n - 2)) \overset{n \gg 1}{\approx} \bar{C}n \propto n \tag{5}$$
$$\text{where} \quad C \equiv (2m - 1)(i + s)/2 \quad \bar{C} \equiv 3m(i + s)$$

where $n$ is the network scale (*i.e.*, $|\mathcal{V}|$), $t$ the task length, $p$ the profile length, $i$ the average instruction length, $s$ the average artifact length, and $m$ the maximum interaction rounds between adjacent agents.

---

[2]Note that although the interaction order is unfolded as a sequence for visualization purposes only, certain sub-topologies (*e.g.*, star) inherently support parallel processing.

[3]Inaccessibility doesn't mean abandonment; when agents incorporate previous contexts into an artifact, these contexts are implicitly embedded and carried forward with the artifact.

[4]The "sink agent" refers to the agent assigned to the sink node. In a multi-sink structure, a final sink node is automatically appended to form a structure with only one sink.

This token complexity analysis implies that, without memory control, context length grows with $n^2$, causing squared increases in time and cost as the network scales. Conversely, our mechanism decouples context length from quadratic to linear growth, effectively suppressing context explosion and enabling better scalability for larger networks.

## 3 EVALUATION

**Baselines**   We select a diverse set of representative methods to facilitate a comprehensive multidimensional comparison:

- CoT (Wei et al., 2022b) is a technically general and empirically powerful approach that endows LLMs with the ability to generate a coherent series of intermediate reasoning steps, naturally leading to the final artifact through process-aware thoughtful thinking.
- AUTOGPT (Richards, 2023) is a versatile agent that employs multi-step planning and tool-augmented reasoning to decompose complex tasks into chained subtasks and leverages external tools within an environment-feedback cycle to progressively develop effective artifacts.
- GPTSWARM (Zhuge et al., 2024) formalizes a swarm of autonomous agents as computational graphs, with nodes as manually-customized functions and edges facilitating information flow, adaptively optimizing node prompts and modifying graph connectivity during collective reasoning.
- AGENTVERSE (Chen et al., 2024d) dynamically assembles and coordinates a team of expert agents in chained or hierarchical structures, employing multi-agent linguistic interaction to autonomously reflect and refine artifacts while displaying emergent social behaviors.

**Datasets and Metrics**   We adopt publicly available and logically challenging benchmarks to evaluate performance across heterogeneous downstream scenarios.

- MMLU (Hendrycks et al., 2021) provides a comprehensive set of logical reasoning assessments across diverse subjects and difficulties, utilizing multiple-option questions to measure general world knowledge and logical inference capabilities. We assess the quality of generated artifacts via *accuracy*, which reflects the correctness of responses to multiple-choice questions.
- HumanEval (Chen et al., 2021), a widely recognized benchmark for function-level code generation, designed for measuring basic programming skills. We assess via *pass@k*, which reflects function correctness across multiple standard test cases.
- SRDD (Qian et al., 2024c) integrates complex textual software requirements from major real-world application platforms, tailored for repository-level software development, involving requirement comprehension, system design, code generation and testing. We assess using the official comprehensive metric encompassing completeness, executability, and consistency.
- CommonGen-Hard (Madaan et al., 2023) tests the ability to generate coherent sentences with discrete concepts, assessing contextual understanding, commonsense reasoning, and creative writing skills. We assess using a comprehensive metric that integrates crucial factors including grammar, fluency, context relevance, and logic consistency (Li et al., 2018).

**Implementation Details**   We construct non-deterministic topologies such as trees and graphs utilizing fundamental structures, including binary trees, layered structures balanced in both width and depth, and random structures crafted by removing edges from a mesh while maintaining connectivity. By default, we employ a topology consisting of approximately four nodes, aligning with multi-agent baselines. GPT-3.5 is employed for interactive reasoning due to its optimal balance of efficacy and efficiency, with each iterative interaction limited to three exchange rounds.

### 3.1 DOES OUR METHOD LEAD TO IMPROVED PERFORMANCE?

We employ the simplest topology—chain—as the default setting for comparative analysis. As demonstrated in Table 1, the chain-structured method consistently surpasses all baselines across most metrics, showing a significant margin of improvement. The primary advantage of MACNET-CHAIN, over a single agent who provides artifacts directly, lies in its facilitation of a procedural thinking in which artifacts are continually reflected and refined. This process effectively mitigates previous inaccuracies or unexpected hallucinations, aligning with previous findings (Cohen et al., 2023; Du

| Method | Paradigm | MMLU | HumanEval | SRDD | CommonGen | Quality |
|--------|----------|------|-----------|------|-----------|---------|
| CoT | 👨 | $0.3544^\dagger$ | $\underline{0.6098}^\dagger$ | $0.7222^\dagger$ | $0.6165^\dagger$ | $0.5757^\dagger$ |
| AUTOGPT | 👨 | $0.4485^\dagger$ | $0.4809^\dagger$ | $0.7353^\dagger$ | $0.5972$ | $0.5655^\dagger$ |
| GPTSWARM | 👥👥 | $0.2368^\dagger$ | $0.4969^\dagger$ | $0.7096^\dagger$ | $0.6222^\dagger$ | $0.5163^\dagger$ |
| AGENTVERSE | 👥👥 | $0.2977^\dagger$ | $\mathbf{0.7256}^\dagger$ | $0.7587^\dagger$ | $0.5399^\dagger$ | $0.5805$ |
| MACNET-CHAIN | 👥👥 | $0.6632$ | $0.3720$ | $\mathbf{0.8056}$ | $0.5903$ | $0.6078$ |
| MACNET-STAR | 👥👥 | $0.4456^\dagger$ | $0.5549^\dagger$ | $0.7679^\dagger$ | $\underline{0.7382}^\dagger$ | $0.6267$ |
| MACNET-TREE | 👥👥 | $0.3421^\dagger$ | $0.4878^\dagger$ | $0.8044$ | $\mathbf{0.7718}^\dagger$ | $0.6015$ |
| MACNET-MESH | 👥👥 | $\underline{0.6825}$ | $0.5122^\dagger$ | $0.7792^\dagger$ | $0.5525^\dagger$ | $\underline{0.6316}^\dagger$ |
| MACNET-LAYER | 👥👥 | $0.2780^\dagger$ | $0.4939^\dagger$ | $0.7623^\dagger$ | $0.7176^\dagger$ | $0.5629^\dagger$ |
| MACNET-RANDOM | 👥👥 | $\mathbf{0.6877}$ | $0.5244^\dagger$ | $\underline{0.8054}$ | $0.5912$ | $\mathbf{0.6522}^\dagger$ |

Table 1: The overall performance of LLM-driven methods across various datasets, including both single-agent (👨) and multi-agent (👥👥) paradigms. Quality represents the average performance over all tasks. For each dataset, the highest scores are highlighted in bold, while the second-highest scores are underlined. A dagger (†) denotes statistically significant differences ($p \leq 0.05$) between a method and our chain-structured setting.

et al., 2024a; Qian et al., 2024b). Moreover, we observe that CoT exhibits strong performance on certain datasets, which is largely because the underlying knowledge of widely-researched benchmarks is already embedded in foundational models, giving single agents a notable capability in these relatively "simple" tasks. While GPTSWARM self-organizes agents through dynamic optimization of nodes and edges, it necessitates extensive task-specific customization for all nodes and edges, complicating usage and thus hindering seamless generalization to heterogeneous downstream tasks. Given the growing need for highly performant and automatic real-world systems, it is impractical to expect that all preparatory knowledge can be fully pre-encoded in foundation models, nor can specific adaptations be pre-made for all unforeseen complex tasks. Fortunately, MACNET bridges this gap by automatically generating various networks through simple hyperparameters (*e.g.,* topology type and scale), enabling agents to engage in cooperative interactions without needing specific adjustments[5], which represents a promising pathway to achieving both autonomy and generalizability. Furthermore, we simulate a regression to graph-of-thought reasoning (Besta et al., 2024a) with a simplified agent by ablating agents' profiles, which led to an average performance drop of 3.67% across all topologies. This result underscores the effectiveness of collective intelligence over singular-aspect reasoning, as the latter represents a variant of dimensionality reduction within multi-agent environments, inevitably blocking its potential to extrapolate potential opportunities.

### 3.2 HOW DO DIFFERENT TOPOLOGIES PERFORM AGAINST EACH OTHER?

To gain a deeper understanding of the impact on organizational structures within multi-agent collaboration, we examine MACNET's topologies across six representative topologies. The analysis focuses on three key perspectives: density, shape, and direction.

**Density Perspective** Table 1 illustrates that different types of topologies vary significantly in effectiveness for specific tasks; no single topology consistently excels across all tasks. For instance, a chain topology is more suitable for software development, while a tree topology is ideal for creative writing. This phenomenon may arise from the inherent suitability of software engineering to a linear process, which is accomplished through sequential steps such as analysis, coding, review, and testing; in contrast, tasks requiring high creativity necessitate more divergent structures to foster agent interactions from various aspects. Additionally, higher interaction density, associated with edge density (see Figure 5), correlates with improved average performance across the three primary topological types. Specifically, the densely connected mesh topology outperforms the moderately dense tree topology, which in turn outperforms the sparsely connected chain topology. This can be

---

[5]Experiments with open-source models demonstrate a similar pattern.

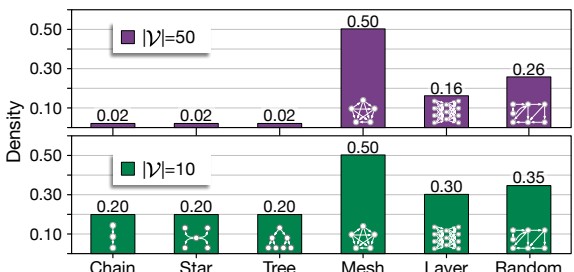
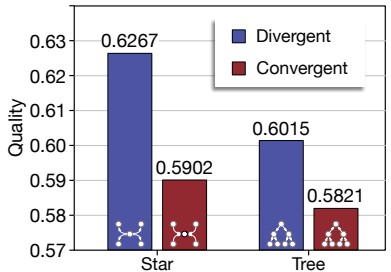

Figure 5: Density of different topologies at different scales.

Figure 6: Comparison between topologies and their reversed counterparts.

attributed to the fact that increased density natually prolongs the reasoning process among collective agents, potentially enhancing opportunities for optimizing artifacts from various aspects.

**Shape Perspective**  Despite the intuitive appeal of densest interactions (*i.e.,* mesh), they do not always yield optimal performance. In contrast, irregular topologies often demonstrate statistically significant advantages. We hypothesize that this phenomenon is because overly dense interactions can overwhelm agents with information overload, impeding effective reflection and refinement. Conversely, network randomization frequently induces small-world properties (Watts & Strogatz, 1998), characterized by a shorter *average path length*[6] or a higher *clustering coefficient*[7]. These random edge connections, akin to residual connections, can link "unacquainted" agents via direct shortcuts, transforming them into "acquaintances" and implicitly reducing the average path length, which naturally decreases the likelihood of long-distance artifact invisibility. This phenomenon, seemingly counterintuitive when compared to well-established regular organizational structures in the real world, suggests that collaboration patterns in an agent's world need not precisely mirror those in human society. Additionally, random topologies consume approximately 51.92% less time than mesh topologies, striking an optimal balance between reduced density and enhanced efficiency, thus serving as a more practical choice. It has also been noticed that, with the same density, star-shaped topologies that are "wider" tend to perform better than "deeper" tree-shaped ones. This is primarily due to the memory control mechanism; while it efficiently manages the spread of overly lengthy contexts across the network, it may cause deeper topologies to lose track of distant agents, occasionally resulting in artifact version rollbacks (Qian et al., 2024a). This points to an empirical search strategy that manages network scale and clustering coefficients, whether through automated searching or manual design, to find an optimal balance between effectiveness and efficiency. Delving deeper, an in-depth inductive bias analysis reveals that in closed-domain scenarios (*e.g.,* logical choices), a chain structure significantly aids in facilitating step-by-step reasoning. Conversely, a proliferation of parallel branches (*e.g.,* stars) can lead to convoluted brainstorming, which may not always be advantageous. In open-domain scenarios, topologies characterized by more convergent nodes are shown to revise artifacts more frequently and produce longer artifacts[8]. This occurs because more convergent nodes, with increased input diversity, increase the likelihood of refining artifacts, benefiting length-sensitive metrics as longer artifacts are more likely to meet rich requirements. Ultimately, no task is confined to a particular topology; the optimal configuration should be chosen based on the openness of scenarios, available computing resources, and associated reasoning costs.

**Direction Perspective**  Beyond density and shape perspectives, the inherent asymmetry in certain topologies—where reversing the edges results in a topologically distinct configuration—has interested us in exploring the effects of reversed topologies. As shown in Figure 6, merely reversing the directions of specific topologies can lead to significant performance degradation. Typically, divergent topologies, characterized by having more child nodes than parent nodes, substantially outperform their convergent counterparts. Intuitively, artifact propagation diverges smoothly, enabling each agent

---

[6]Average path length (Albert & Barabasi, 2002) is the average number of steps along the shortest paths for all possible pairs of network nodes, which is a measure of the efficiency of information transport on a network.

[7]The clustering coefficient measures the connectivity density among a node's neighbors (Strogatz, 2001).

[8]The layer topologies exhibit a 92.16% modification probability and an average artifact length of 586.57, compared to 68.48% and 308.26 for chain topologies

to discuss artifacts from varied aspects. In contrast, aggregating multiple artifacts at a convergent node is more challenging, highlighting the complexity of integrating diverse aspects into a cohesive artifact. Therefore, to minimize potential degradation during artifact aggregation, it is recommended to employ topologies that maximize divergence while minimizing convergence.

### 3.3 COULD A COLLABORATIVE SCALING LAW BE OBSERVED?

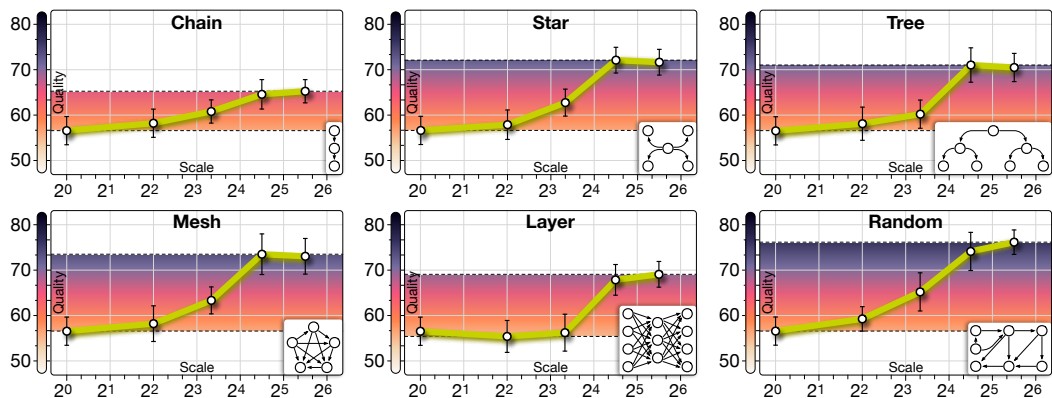

Figure 7: Scaling performance of multi-agent collaboration under different topologies. Quality represents the average performance over all tasks.

**Trend Perspective** Recall the neural scaling law, which posits that increasing neurons leads to an continual performance improvement (Kaplan et al., 2020). To investigate the *collaborative scaling law*, which excavates the relationship between agent scale and performance, we initiated an attempt by exponentially increasing the number of nodes ($|\mathcal{V}|$) from $2^0$ (regressing to a single-agent variant) to $2^6$ (equating to over a thousand agents in a mesh network). As depicted in Figure 7, scaling our networks initially grows slowly in the quality of artifacts generated by various multi-agent systems, then leads to a rapid improvement before reaching a saturation point. This pattern resembles a sigmoid-variant function:

$$f(|\mathcal{V}|) = \frac{\gamma}{1 + e^{-\beta(\log |\mathcal{V}| - \alpha)}} + \delta \tag{6}$$

where $\{\alpha, \beta, \gamma, \delta\}$ are real numbers specific to a particular topology. Roughly speaking, a node magnitude of $2^4$ appears to be a reasonable choice. However, considering the efficiency of sparse topologies and the superior performance of dense ones, we advocate balancing shape and scale through multidimensional trade-offs when applying this trend to various downstream applications. This finding suggests that many existing agent systems may be operating below their full potential, which underscores a promising path for enhancing performance by increasing the number of agents, provided they collaborate effectively, rather than solely focusing on scaling foundational models.[9]

Besides, the validation of baseline scaling reveals that equalizing the number of LLM calls—whether through majority voting in closed-domain tasks (Chen et al., 2024b) or best-of-N in open-domain tasks (Sessa et al., 2024)—consistently highlights a lack of effective scalability across all baselines. Majority voting enhances performance by merely 0.9%, even when augmented with CoT or AUTO-GPT, plateauing at approximately eight agents. AGENTVERSE implicitly reduces to a star topology and frequently encounters context explosion issues when scaling beyond thirty agents, thus hindering scalability. The energy-intensive setup of GPTSWARM necessitates manual, task-specific structuring and prompting, which restricts both multitasking capabilities and overall scalability.

**Timing Perspective** The neural scaling law requires models with at least a billion parameters and over $10^{22}$ training FLOPs to show emergent trends (Schaeffer et al., 2024). In contrast, collaborative emergence in MACNET manifests at much smaller scales, with most topologies reaching performance

---

[9]Looking further, this fitting only reflects a general pattern from the perspective of network scales; future research should aim for a more precise characterization by incorporating additional factors like profiles, tools and communication protocols, or social routing.

saturation with approximately a hundred agents. The fundamental reason is that neuron coordination (during training) relies on numeric matrix operations, requiring all neurons to precisely and simultaneously learn from scratch to assimilate extensive world knowledge. Conversely, individual agents (during inference) already possess certain knowledge from the foundational models, and their coordination through interdependent interactions utilizes existing reasoning skills to disseminate knowledge from diverse aspects; the most critical aspects for artifact refinement in agents' interactions typically do not require such a large scale to be thoroughly reflected and refined. Thus, alongside neuron collaboration, agent collaboration may serve as a "shortcut" to enhance intelligence levels, especially when large-scale retraining resources such as data and hardware are constrained.

### 3.4 WHAT FACTORS MIGHT CONTRIBUTE TO COLLABORATIVE EMERGENCE?

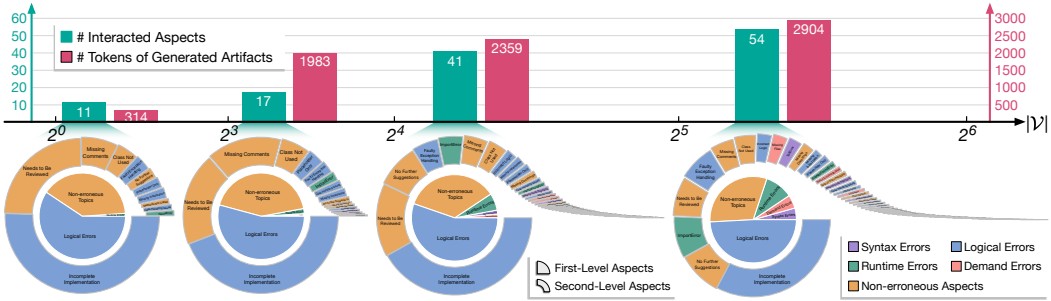

Figure 8: The number and distribution of aspects in agent interactions, along with the length of artifacts. The pie chart features primary aspects in the inner circle and secondary aspects in the outer circle, with a long-tail layout to visualize tail aspects. Zoom in for more detailed information.

To delve deeper into the underlying mechanisms, we selected the moderately-dense layer typology employed in software development, which serves as a representative case, with similar phenomena consistently occurring in other topologies and scenarios. Specifically, we classified the aspects discussed in agents' interactions into five main categories (Oh & Oh, 2022; Kohn, 2019): four levels of errors (syntax, runtime, logic, and unmet requirements) and a non-error category; each category contains multiple subcategories. Figure 8 displays the total number of interaction aspects, along with their detailed distribution. Within smaller topologies ($2^0 \le |\mathcal{V}| \le 2^3$), the limited interaction density confines aspects to approximately a dozen secondary aspects. However, as the network expands ($2^4 \le |\mathcal{V}| \le 2^6$), the interaction density increases quadratically, resulting in a sudden increase to dozens of aspects, followed by a more gradual rise. This progression closely parallels the trend observed in emergent capabilities, which may partially attribute the emergence to the sharp rise in detailed interacted aspects among agents. This phenomenon occurs because the token distribution from underlying models typically follows a long-tail pattern, necessitating larger-scale sampling to likely capture these tail tokens. Consequently, this encourages the emergence of more infrequent "tail aspects", allowing the collaborative process to extend beyond the most common aspects. Theoretically, the probability of a long-tail token $t$ appearing at least once in $n$ samples is:

$$p^n(t) = 1 - (1 - p(t))^n \propto 1 - (1 - 1/r(t))^{|\mathcal{V}|^2} \quad \lim_{|\mathcal{V}| \to \infty} p^n(t) = \lim_{n \to \infty} p^n(t) = 1 \quad (7)$$

where $p(t) \propto 1/r(t)$ represents a standard Zipf's law characterizing a long-tailed distribution (Newman, 2005); the sampling size $n$ is proportional to the interaction density, *i.e.,* $n \propto |\mathcal{V}|^2$. It can be inferred that increasing the network size significantly enhances the probability of tail token occurrences, gradually approaching an asymptote. This probability becomes an inevitable event once the sample size is sufficiently large. Statistically, when a critic suggests a particular aspect, there is a 93.10% statistical likelihood that an actor will implement the recommended refinement rather than disregard it. The scaling up enables critics to pinpoint finer issues within artifacts, guiding actors to initiate corresponding refinements. Consequently, each round of dialogue in the collaborative process refines artifacts from different aspects, naturally elevating the probability of producing more nuanced artifacts (Liang et al., 2024; Du et al., 2024a; Cohen et al., 2023).

In response to multidimensional considerations, scaling agents accordingly prolongs the overall length of artifacts. For instance, the token length increased by 7.51 times when scaling from $2^0$ to

$2^4$. This characteristic, over small-scale networks, facilitates the integration of detailed requirements, performance optimization, and other advanced factors, potentially encompassing abilities that shorter artifacts cannot. This is mainly due to the graph's naturally divergent and convergent topologies, which enable artifacts to porpagate for strength-aggregated refinement. Therefore, unlike majority voting, this paradigm fosters interdependent interaction and length-extended regeneration among diversified artifacts, thereby producing more comprehensive ones.

## 4 RELATED WORK

**Large Language Models** Trained on vast datasets through next token prediction (Vaswani et al., 2017) and capable of manipulating billions of parameters (Muennighoff et al., 2024), LLMs have become pivotal in natural language processing due to their seamless integration of extensive knowledge (Brown et al., 2020; Bubeck et al., 2023; Radford et al., 2019; Touvron et al., 2023; Wei et al., 2022a; Shanahan et al., 2023; Chen et al., 2021; Brants et al., 2007; Chen et al., 2021; Ouyang et al., 2022; Yang et al., 2024; Qin et al., 2024b). Central to this breakthrough is the neural scaling law, which posits that loss descends as a power law with model size, dataset size, and the amount of compute used for training (Kaplan et al., 2020; Smith et al., 2022; Ruan et al., 2024). The principle underscores that scaling up language models can lead to emergent abilities—where performance experiences a sudden leap as the model scales (Wei et al., 2022a; Schaeffer et al., 2024).

**Autonomous Agents** Despite these advancements, LLMs possess inherent limitations in enclosed reasoning, driving further research to integrate advanced capabilities such as context-aware memory (Park et al., 2023; Hua et al., 2023), tool use (Schick et al., 2023; Qin et al., 2024a), procedural planning (Wang et al., 2023a; Zelikman et al., 2024), and role playing (Chan et al., 2024; Wang et al., 2024c; Liu et al., 2024a), thereby transforming fundamental LLMs into versatile autonomous agents (Richards, 2023; Shinn et al., 2024; Zhao et al., 2024; Lin et al., 2023; Mei et al., 2024; Chu et al., 2024). Along this line, multi-agent collaboration has proven beneficial in uniting the expertise of diverse agents for autonomous task-solving (Khan et al., 2024; Liang et al., 2024; Qian et al., 2024c; Wang et al., 2024b;a; Zhou et al., 2024; Talebirad & Nadiri, 2023; Chen et al., 2024c; Li et al., 2023b), which has widely propelled progress across various domains such as software development (Hong et al., 2024; Qian et al., 2024a), game playing (Vinyals et al., 2019), personalized recommendation (Wang et al., 2023b; Zhang et al., 2023), medical treatment (Tang et al., 2023; Li et al., 2024a), financial marketing (Gao et al., 2024; Li et al., 2024c), educational teaching (Zhang et al., 2024c; Yu et al., 2024), scientific research (Zeng et al., 2024; Baek et al., 2024; Ghafarollahi & Buehler, 2024) and embodied control (Guo et al., 2024; Chen et al., 2024f; Mandi et al., 2023). Technically, in contrast to straightforward majority voting where individuals act independently (Chen et al., 2024b), collective emergence (Woolley et al., 2010; Hopfield, 1982; Watts & Strogatz, 1998) posits that effective collaboration should evolve into an integrated system that promotes interdependent interactions and thoughtful decision-making (Li et al., 2024b; Piatti et al., 2024). As such, recent studies differentiate agents into distinct expertise and encourage task-oriented interactions, forming a chained workflow to sequentially reach final artifacts (Qian et al., 2024c). Subsequent research seeks to organize expert agents in a tree structure for hierarchical information propagation (Chen et al., 2024d) or in a graph with predefined node and edge functions (Zhuge et al., 2024).

## 5 CONCLUSION

This study explores the impact of scaling multi-agent collaboration by introducing MACNET, a scalable framework that utilizes graphs to organize agents and orchestrate their reasoning for autonomous task solving. Extensive evaluations reveal that it effectively supports collaboration among over a thousand agents, with irregular topologies outperforming regular ones. We also identify a *collaborative scaling law*—the overall performance follows a logistic growth pattern as agents scale, with collaborative emergence occurring earlier than previously observed neural emergence. We speculate this may be because scaling agents catalyzes their multidimensional considerations during interactive reflection and refinement, thereby producing more comprehensive artifacts. However, our research also indicates that there are limits on the scaling horizon. By extrapolating traditional scaling from training to inference, we posit that agent collaboration could serve as a "shortcut" to bypass the need for resource-intensive retraining by employing inference-time procedural thinking.

ACKNOWLEDGEMENTS

The work was supported by the Tencent Rhino-Bird Focused Research Program and the Postdoctoral Fellowship Program of CPSF under Grant Number GZB20230348.

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
