# OpenReview forum: "Scaling Large Language Model-based Multi-Agent Collaboration"
_ICLR.cc/2025/Conference — ICLR 2025 Poster_

### Official Review · Reviewer_P62L · 2024-11-03

**Soundness:** 3
**Presentation:** 3
**Contribution:** 3
**Rating:** 8
**Confidence:** 3

**Summary:**

The paper proposes MacNet, a multi-agent collaboration network. Inspired by neural scaling laws, the authors investigate how the number of agents and the communication topology between agents impacts performance when agents are made to collaborate on a given problem. Specifically:

- MacNet creates a Directed Acyclic Graph (DAG) of agents that attempt to solve a given task in analogy to a neural network. The **nodes and edges are agents**, where the nodes are "executors", which accomplish tasks, while the edges are "instructors", which evaluate the task and provide feedback. Agents are allowed to **interactively and iteratively refine** their subtask solutions, where an instructor asks an executor to refine their proposed subtask solution. To avoid **content explosion**, nodes in the network are expected to only pass on the proposed solution, not its entire reasoning, and they are also expected to aggregate incoming solutions. Importantly, MacNet does not have a specific topology; the method can be applied to many different types of topologies, including random topologies.

- To evaluate the performance of MacNet, the authors evaluate on MMLU, HumanEval, SRDD and CommonGen-Hard. They compare MacNet variants based on different DAG topologies against Chain of Thought, AutoGPT, GPTSwarm and AgentVerse. Their method exhibits significantly better performance on MMLU, mostly worse performance on HumanEval, slightly better on SRDD and sometimes better performance on CommonGen.
  - They compare the performance of different topologies against each other, finding that random topologies often perform statistically better.
  - They observe the performance of different topologies across multiple network sizes, demonstrating a non-decreasing performance trend.
  - They perform an error analysis on one of the topologies at different sizes, finding longer tails when the number of agents increases which diversifies the solutions, making the proposed solutions more comprehensive.

Ultimately, they conclude that agent collaboration could bypass the need for retraining by enabling inference time performance scaling through scaling multi-agent collaboration.

**Strengths:**

# originality

* The authors study the scaling of the number of agents during inference, a relatively **novel, unexplored problem**.

* The authors **extend the interaction DAG setting** to a functionally bipartite structure with agents as nodes and edges, and **present representative topologies** that capture prevalent DAG structures

* The authors **address the limitations of prior work** by avoiding content explosion

# quality

* The authors describe the problem setting with the appropriate technical language, as well as **motivating theoretical justifications** for MacNet's advantages over existing solutions.

* The authors compare against a set of **timely and relevant baselines** on ** diverse datasets**.

* The authors use **identical and reproducible settings** across methods.

* The authors provide **multiple perspectives** on their results, connecting observed phenomena to novel and documented insights.

# clarity

* The paper is **well-structured**, making it easy to read and follow along.

* The paper uses **interesting visualizations** to communicate interesting insights, such as longer tails with more agents.

* The paper employs **precise formal language and notation** to effectively communicate their points.

# significance

* The paper has a significant impact on **LLM-based applications**

* The paper is very **timely** as the research on multi-agent systems begins to regain the interest of the machine learning community.

* The paper offers **interesting insights** into collaborative inference-time structures across many agents, which is relevant to ongoing discussions around agent proliferation.

* The paper outlines **future directions of work**.

**Weaknesses:**

# Problem Setting

* **collaborative emergence is undefined**, even though it seems to be a central concept of the paper.

# Results

## Major
- **Lack of transparent methods**: The use of the closed source line of GPT models hinders the analysis. The authors take the stance that the multi-agent scaling laws are analogous to the neural scaling laws, but we do not know much about the underlying models. It would be more judicious to compare open-source models, for which we know the FLOPs and parameter counts. This could allow for comparisons such as a single 7B model vs a MacNet of 7 1B models. An additional point here is that we do not know what type of post-training the GPT models undergo, which could help in the collaboration. The main point of this criticism is the following: **the use of a closed-source method in your analysis effectively hinders the potential for your results to generalize to other LLM-based agents**. We don't know the impact of model size, architecture or training strategy for the GPT models, so we can't know how the established analysis and scaling laws transfer to other architectures.


- **Entangled sources of improved performance**: Since the actual FLOPs used are not kept track of, it becomes impossible to tell whether the improved results are due to additional computational resources spent on figuring out a solution. It would be desirable to **isolate the impact of the multi-agent collaboration** from other factors, such as how much compute is poured into a given solution. Otherwise, it could be that a given MacNet works better because it pours more resources and time into a solution. While this result would be interesting in itself, it would shift the significance of the approach mostly onto methods for scaling inference-time compute, at which point you'd have to normalize for how much time and compute is spent on a given solution. You would then also have to compare against other non-multi-agent methods for scaling inference time compute. Instead, the diversity and comprehensiveness of the solutions output by a MacNet could be the subject of a greater analysis, while being more direct about the resource requirements.

## Minor
- **Lack of scaling law analysis**: The scaling laws, although the sigmoid functional form is described, are not actually fit, nor analyzed. Furthermore, there is little to no discussion as to *why* the scaling laws should take a sigmoidal form in this case.


# Overall

The presentation of the paper is solid. The writing is easy to understand and the use of technical language is judiciously sparse. The problem setting is good and the contribution would be too, but the soundness of the claims is affected by the limits to the results outlined above. This unfortunately impacts the value of the contribution itself, as the results are not wide-ranging.

**Questions:**

- Doesn't the interative feedback between instructors and executors make the graph not directed? You could get stuck in a loop there.

- Have you looked into letting the agents attempt to determine what a good topology is? I.e. Agents specify to which other agent they believe they should send their result.

- If quality is the average across all tasks, does each benchmark have the same amount of tasks? Or did you average specifically task-wise, or across all 4 datasets?

---

### Official Review · Reviewer_e6WL · 2024-11-04

**Soundness:** 4
**Presentation:** 4
**Contribution:** 4
**Rating:** 8
**Confidence:** 3

**Summary:**

This paper presents pioneering work on scaling large language model-based multi-agent collaboration to the level of a thousand agents. The authors propose a method to automatically organize LLM agents into a directed acyclic graph (DAG) without needing specific adjustments for generalization. Comprehensive evaluations are conducted on diverse benchmarks against a variety of baselines. The influence of topologies, the number of agents, and other factors on collaboration performance is discussed in depth. The paper is well-written with a clear flow. The scaling law experiments involving over a thousand agents represent a truly rare finding in the literature I have read about LLM agents. I would highly recommend accepting this paper.

**Strengths:**

The scaling of multi-agent systems powered by LLMs is a significant research question for the community.

The idea of organizing LLM agents using a DAG is not novel; however, the authors present a comprehensive analysis of factors including types of topologies, densities, directions, and scales.

The experiments involve at maximum of 1000 LLM agents, with the help of the proposed memory control mechanism to avoid context explosion.

The proposed method is highly flexible and generalizable, requiring little human supervision or prior knowledge as claimed by the authors.

**Weaknesses:**

What is the rationale for using "instructor" for edges and "executor" for nodes? It seems that both can be treated as nodes with different profiles, with edges between them indicating interactions.

Implementation details are somewhat brief. As mentioned in lines 259-261, agent profiles and tools are generated by GPT-4 and randomly assigned to network agents. It is unclear how this process can be fully automated without interventions or adjustments at the scale of 1,000+ agents. Is there any guarantee that the randomly generated agent network can solve the given problem?

How can we ensure that the memory control mechanism does not omit context information? Additionally, how many agent profiles and tools are generated?

How many repeated measurements are conducted for each method across different datasets? What is the variance in performance?

**Questions:**

Questions are asked in above sections.

---

### Official Review · Reviewer_2D7q · 2024-11-04

**Soundness:** 3
**Presentation:** 4
**Contribution:** 3
**Rating:** 6
**Confidence:** 4

**Summary:**

The paper explores the scaling law of collaboration for LLM-based agents. The authors propose a paradigm, MacNet, that allows for the collaboration of up to thousands of agents, and demonstrate its superiority over baseline methods across four diverse datasets. Utilizing MacNet, the authors investigate the impact of different topological structures on multi-agent collaboration, distill a scaling law concerning the number of nodes in multi-agent graphs, and analyze the factors leading to the emergence of multi-agent collaboration through case studies.

**Strengths:**

1. The paper is clearly articulated and well-structured, with the discussion on "why random topologies perform better" and "tail aspects being factors leading to the emergence of collaboration" being particularly enlightening.
2. The experimental section of the paper is comprehensive (although there may be areas lacking in rigor), covering a wide range of perspectives in its design. I believe the work presented in this paper can inspire the entire research community.

**Weaknesses:**

As an empirical research paper, the experimental design and conclusions of the paper are particularly important and need to be sufficiently rigorous. I currently have some concerns.

(Major) 1. Regarding the emergence in multi-agent systems. The paper discusses the laws of neural scale emergence and agent collaboration emergence between lines 427 and 429. However, the claim of collaboration emergence might not be rigorous enough. For instance, in neural emergence, a neural network model can evolve from merely recognizing images to handling PhD-level problems, which is a significant leap. In contrast, the emergence described in this paper only involves an improvement in success rate from around 55% to 70% on certain datasets, a particularly small span that might not even qualify as emergence. If an appropriate case could be found where multiple agents attempt to solve a very difficult problem, and the agents manage to solve this problem once a certain scale is reached (analogous to human society solving a very complex and challenging problem), it would greatly enhance the paper.

(Major) 2. Corresponding to the above is the selection of benchmarks and the design of metrics. A fundamental question is why, even with the use of up to thousands of agents, the performance improvement is only around 20%, without a breakthrough improvement akin to advancing from grade school level to PhD level. Another phenomenon is that, as mentioned in section 3.4 of the paper, while the number of Interacted Aspects continues to increase when the number of agents ranges from 2^5 to 2^6, there is no significant improvement in performance. Does this imply that the current tasks are not challenging enough? Are the metrics for quantifying performance not rich enough?

(Minor) The paper mentions " alongside neuron collaboration, agent collaboration may serve as a shortcut to enhance intelligence levels, especially when large-scale retraining resources such as data and hardware are constrained." To arrive at this general conclusion, conducting experiments only on the same series of large models (GPT3.5 and GPT4) might be insufficient. Experiments with other types of LLMs might be necessary.

**Questions:**

Beyond the major concerns I list, there are the following questions:
1. Can the authors provide mathematical insights regarding the implications of multiple LLMs collaborating? From a mathematical theory standpoint, what is the difference between increasing the size of large models and increasing the number of large models collaborating?
2. The abstract part mentions "with collaborative emergence occurring earlier than traditional neural emergence". How should this be understood? Can the authors provide a detailed discussion?

---

> ### Comment · Reviewer_2D7q · 2024-11-25
>
> Thank you for providing clarifications on the previously imprecise statements regarding scaling emergence, conducting experiments on a broader variety of LLMs, and elucidating some of the qualitative conclusions that were previously less accurate. I concur with the authors' perspective:
> "Therefore, in more complex datasets where the current model's capability is less than 5%, we anticipate that the contrast in performance improvement would be even more pronounced."
> As the authors have also mentioned, should the related work be conducted on datasets of sufficient complexity and yield similarly insightful conclusions (not necessarily strictly sigmoid curves), I would undoubtedly rate it an 8 or higher. Given my initial score was a 6 (the highest I could offer at that time), **I would have been inclined to adjust my score to a 7**. However, in the absence of a 7 option and my score leaning more towards a 6, I will maintain my original score.
>
> Overall, the authors have accomplished commendable work, and I believe the timing of its presentation at this conference aligns well with the AI field's current interest in exploring the collaborative laws of multiple LLMs. I wish the authors all the best.

---

### Official Review · Reviewer_TcPL · 2024-11-04

**Soundness:** 4
**Presentation:** 4
**Contribution:** 3
**Rating:** 6
**Confidence:** 4

**Summary:**

The paper introduces a novel multi-agent collaboration framework, MACNET, which utilizes directed acyclic graphs (DAGs) to organize agents and orchestrate their reasoning for autonomous task solving. In their extensive evaluations, the paper finds that multi-agent collaboration can achieve better performance than the individure agent and also reveals a collaborative scaling law, indicating that performance follows a logistic growth pattern as the number of agents increases. The study also demonstrates the superior performance of irregular topologies over regular ones in multi-agent collaboration, attributing this to the catalyst effect of scaling agents on multidimensional considerations during interactive reflection and refinement.

**Strengths:**

- The paper introduces a pioneering framework that employs directed acyclic graphs (DAGs) to organize agents within a multi-agent system. It provides insightful comparisons between various topologies.

- The research delves into the impact of scaling the number of collaborative agents on overall performance, uncovering a collaborative scaling law. This law suggests that performance follows a logistic growth pattern as agents are added, which is a significant finding as it parallels the neural scaling law but applies to the domain of agent collaboration.

**Weaknesses:**

Check in Questions.

**Questions:**

I find the modeling of MAS LLM agent and the study of the scaling law in the paper quite intriguing, but I have reservations about the problems that the multi-agent is used to solve.

In the experiments within the paper, single-agent methods like CoT and AUTOGPT are compared, but to my knowledge, for datasets like MMLU, the pure LLMs (without designing agents deliberately) can already achieve a performance of 0.8 as can be check in [paperswithcode](https://paperswithcode.com/sota/multi-task-language-understanding-on-mmlu), and for HumanEval, based on [1],[2], the performance of above 0.8 can also be achieved.  The works I mentioned above are all better than the baselines and this method MACNET.

So, if using a multi-agent collaboration work, considering ten to thousands of agents, only achieves the performance shown in Table 1, I don't think it proves the necessity of multi-agent; in fact, it could be seen as undermining their necessity.

- Compare MACNET against these state-of-the-art single-agent methods and Explain why MACNET underperforms compared to these methods, if that is indeed the case.
- Can you discuss specific scenarios or types of tasks where multi-agent approaches might offer unique advantages over single-agent methods?

[1] Zhong, Li, Zilong Wang, and Jingbo Shang. "Ldb: A large language model debugger via verifying runtime execution step-by-step." arXiv preprint arXiv:2402.16906 (2024).

[2] Shi, Yuling, et al. "From Code to Correctness: Closing the Last Mile of Code Generation with Hierarchical Debugging." arXiv preprint arXiv:2410.01215 (2024).

---

### Meta-Review · Area_Chair_bVPW · 2024-12-19

**Metareview:**

This paper, using the MACNET framework, investigates the impact of scaling the number of collaborative agents on overall performance, revealing a collaborative scaling law. However, the experimental setup is significantly confusing, particularly in the implementation details, including agent profiles and tools, and the objective reporting of performance. The overall framework and idea of the paper are relatively simple and easy to understand, and the authors have effectively articulated their core contributions.

**Additional Comments On Reviewer Discussion:**

To ensure fairness and impartiality, the Area Chair personally reviewed the paper when making the decision, focusing on the core contributions of the work and closely examining the rebuttal between the authors and reviewers. The interactions during the rebuttal period were thorough and aligned with ICLR’s expectations for the review process. While the paper demonstrates some shortcomings in terms of the objective presentation of experimental results and the logical explanation behind the framework, the reviewers’ feedback was carefully considered. Based on these factors, the paper has been accepted as a Poster for ICLR 2024.

---

### Decision · Program_Chairs · 2025-01-22

Accept (Poster)